# Prevalence of tobacco dependence and associated factors among patients with schizophrenia attending their treatments at southwest Ethiopia; hospital-based cross-sectional study

**Defaru Desalegn** * , **Zakir Abdu, Mohammedamin Hajure**

Department of Psychiatry, College of Health Sciences, Mettu University, Mettu, Oromia Region, Ethiopia

* defdesalegn2007@gmail.com

## Abstract

### Background

Tobacco smoking is the most typically employed in patients with mental disorders; among them, patients with schizophrenia are the very best users. The rate of smoking among patients with schizophrenia is between two and three times greater than the general population in western countries. However, there is a scarcity of studies on the magnitude and associated factors of tobacco dependence among patients with schizophrenia in Ethiopia. Therefore, we assessed the prevalence of tobacco dependence and associated factors among patients with schizophrenia at Mettu Karl referral, Bedelle, and Agaro hospitals, Southwest, Ethiopia.

### Method

Hospital-based the multistage stratified cross-sectional study design was conducted among 524 patients with schizophrenia who are on treatment. Fagerstrom Test for Nicotine Dependence (FTND) was used to screen the prevalence of tobacco dependence. Analysis of data was done using SPSS version 24.

### Result

The prevalence of tobacco dependence among study participants was 22.3% (95% CI) (18.6, 26). Concerning the severity of tobacco dependence, 3.5%, 13.8%, and 5% of the respondents report moderate, high, and very high levels of tobacco dependence respectively. The proportions of tobacco dependence among male schizophrenic patients 88 (25.8%) were higher compared to their counterparts 27 (15.5%). After controlling the effects of cofounders in the final regression analysis, male gender (AOR 2.19, 95% CI = 1.25, 3.83), being on treatment for more than 5years (AOR 4.37, 95% CI = 2.11, 9.02), having a history of admission (AOR 4.01, 95% CI = 1.99, 8.11), and family history of mental illness

**Data Availability Statement:** The data underlying this study may not be released publicly, as participants did not consent to public sharing of the data. Interested, qualified researchers can access

the data by sending requests to Desalegn Chilo (Dean of college of health science, Mettu University) at desalegn.chilo@meu.edu.et.

**Funding:** The study was funded by Mettu University. The university had no role in the design of the study, in the collection, analysis, and interpretation of the data; or in writing the manuscript.

**Competing interests:** The authors declare that they have no competing interests.

**Abbreviations:** AUD, Alcohol Use Disorder; CAGE, Cut down, Annoyed, Guilty, an Eyeopener; CI, Confidence Interval; FTND, Fagerstrom Test for Nicotine Dependence; LMICs, Low- and Middle-Income Countries; OR, Odds Ratio; SPSS, Statistical Package for Social Science; UK, United Kingdom; WHO, World Health Organization.

(AOR 1.90, 95% CI = 1.04, 3.48) were shown to have a significant positive association with tobacco dependence.

## Conclusion and recommendation

A study show a significant proportion of tobacco dependence among people living with schizophrenia. Factors like, being male gender, being on treatment for more than 5 years, having a history of admission, and family history of mental illness was found to have a significant positive association with tobacco dependence. Hence, there is a need for coordinated and comprehensive management clinically to manage tobacco dependence along with identified risk factors in patients with schizophrenia. Also the finding call for the clinicians, managers, ministry of health and other stakeholders on the substance use prevention strategies that target personal and environmental control.

## Introduction

Globally, cigarette smoking is among the highest 5 causes of risk mortality and is that the single largest preventable reason behind death; it promotes quite five million annual deaths, inflicting 11% of ischemic heart deaths and quite seventieth of respiratory organ, cartilaginous tube, and trachea cancer [1]. According to the report of WHO, cigarette smoking in the developed countries is the cause of 20% of preventable death [2]. The report shows that tobacco is estimated to kill about one billion people in the 21st century, particularly from low- and middle-income countries (LMICs) [3].

People living with mental illness are more likely to smoke and be at greater risk for smoking-related health problems than the general population [4]. Mortality from smoking is higher among individuals living with mental illness supported by the report of the study comprised 600,000 respondents where tobacco-related conditions were contributed to 53% of total deaths in schizophrenia [5]. Smoking in schizophrenic patients contributes to a 20% decrease in their life expectancy compared to members of the general population [6].

The impact of smoking among patients with schizophrenia not only increases metabolism and vascular risks [7], also increases suicide risk [8]. It decreases the antipsychotic therapeutic effects as smoking induce the medication metabolism in the liver reducing up to 48% of the active metabolites in serum [9]. Schizophrenic-smokers show more hospitalization frequency (than schizophrenic non-smokers) and also require more depot medication, having fewer adherences to treatment [10].

A meta-analysis of 42 epidemiological studies across 20 different countries showed that people with schizophrenia have more than five times the odds of current smoking than the general population and smoking cessation rates are much lower in smokers with schizophrenia compared with the general population [11]. Another meta-analysis study was done in 8 countries based on 14 studies found that the average prevalence of current smokers among male schizophrenia patients were 72% [12].

According to one study from the United Kingdom (UK) done among 8 million patients, the prevalence of smoking among psychotic patients (schizophrenia, schizotypal and delusional disorders) was 44.6% [13].

A cross-sectional study was done in China among inpatient schizophrenic patients found that the prevalence rate of current smoking was 40.6%, which was 57.5% in males and 6.3% in females and the study described that factors such as being male sex, older age, poor marital

status, alcohol use, use of first-generation antipsychotics, longer duration of illness, more fre-
quent hospitalizations, and more severe negative symptoms were independently associated
with current smoking [14].

A cross-sectional study was done in Singapore among male schizophrenic patients found
that the lifetime prevalence of smoking cigarettes and current smokers are 54.1% and 42.4%
respectively [15], in Iran as high as 71.6% [16], in Scotland 53.4% [17], and in Turkey 49%
[18].

One a cross-sectional descriptive study was done in Southwest Nigeria among 367 patients
with schizophrenia found that the lifetime prevalence and a current smoking rate of 20.4% and
25.9% were reported respectively [19]. Facility-based a cross-sectional study done in Jimma
medical center in Ethiopia on tobacco dependence among people with mental illness found
that the prevalence of current tobacco dependence among the study participants is 18.5% and
specifically the prevalence of tobacco dependence among patients with schizophrenia was
29.1%; furthermore, the study described their level of tobacco dependence as 57.7% moderate,
29.5% higher and 12.8% very high [20].

Little information is available regarding nicotine dependence among patients with schizo-
phrenia in Ethiopia. Thus, this study was aimed to assess the prevalence of tobacco dependence
and associated factors among patients with schizophrenia attending their treatments at Mettu
Karl, Bedelle, and Agaro hospitals, Southwest Ethiopia.

## Materials and methods

### Study setting and period

The study was conducted from 1st April to 30th June 2019 at the psychiatric clinic of three gov-
ernmental health institutions (Hospitals) found in southwest Ethiopia, namely Mettu Karl
referral, Bedelle and Agaro hospitals, which were 600 kilometer, 426 kilometer, and 397 kilo-
meter far from Addis Ababa to the southwest, the capital city of Ethiopia, respectively.

### Study design

Hospital-based a cross-sectional study was conducted.

### Source population

All patients with schizophrenia attending follow-up treatments at Mettu Karl referral, Bedelle,
and Agaro hospitals psychiatric clinic

### Study population

Sample of patients with schizophrenia who attended the outpatient treatment at the psychiatric
clinic of Mettu Karl referral, Bedelle, and Agaro hospitals during the data collection period

**Inclusion and exclusion criteria.** Adult patients (aged 18 and above) with schizophrenia
who were already diagnosed previously as per the diagnostic criteria of the Diagnostic Statisti-
cal Manual of Mental Disorders, 4th and or 5th edition (DSM-IV or DSM-V) were included in
the study and patients with schizophrenia whose illness was in the acute stage or in exacerba-
tion of symptoms were excluded from the study.

### Sample size determination

The minimum the sample size required for this study was determined by using the formula to
estimate single population proportion, $n = ((z_{\alpha/2})^2 p(1-p))/d^2$ by using the following assump-
tions: the prevalence of tobacco dependence among patients with schizophrenia at the Jimma

medical center was 29.1% [20], a 95% confidence interval (CI), 5% the margin of error and a non-response rate of 10%. We applied the single population proportion formula to give in = $(1.96)^2 * 0.291 (1–0.291) / (0.05)^2 = 317$.

Since multistage stratified sampling the technique was used to select study participants, using design effect the sample size was multiplied by 1.5, giving 476 considering that the questionnaire was self-administered and finally adding a 10% non-response rate, the final number of the study subject became 524.

## Sampling technique

The multi-stage stratified sampling technique was used to select the study participants. Stratification was first done on the zone level, then by the hospitals found in the zones (**Fig 1**).

## Data collection procedure and tools

An interviewer-administered a structured questionnaire was used to collect information. Questionnaires about demographic and other clinical factors were developed after an extensive review of the literature and similar study tools. We employed nine interviewers' (data collectors) for 2 months data collection period for collecting data from the participants (patients with schizophrenia). The interviewers' background or expertise was that they were all bachelors of degree and master of degree holders in psychiatric nursing. Hence, we (authors) believe that the interviewers' background or expertise (being psychiatric professionals) can lead them to determine the capacity of patients to provide consent for the study. The study was done among patients with schizophrenia who were already diagnosed previously as per the

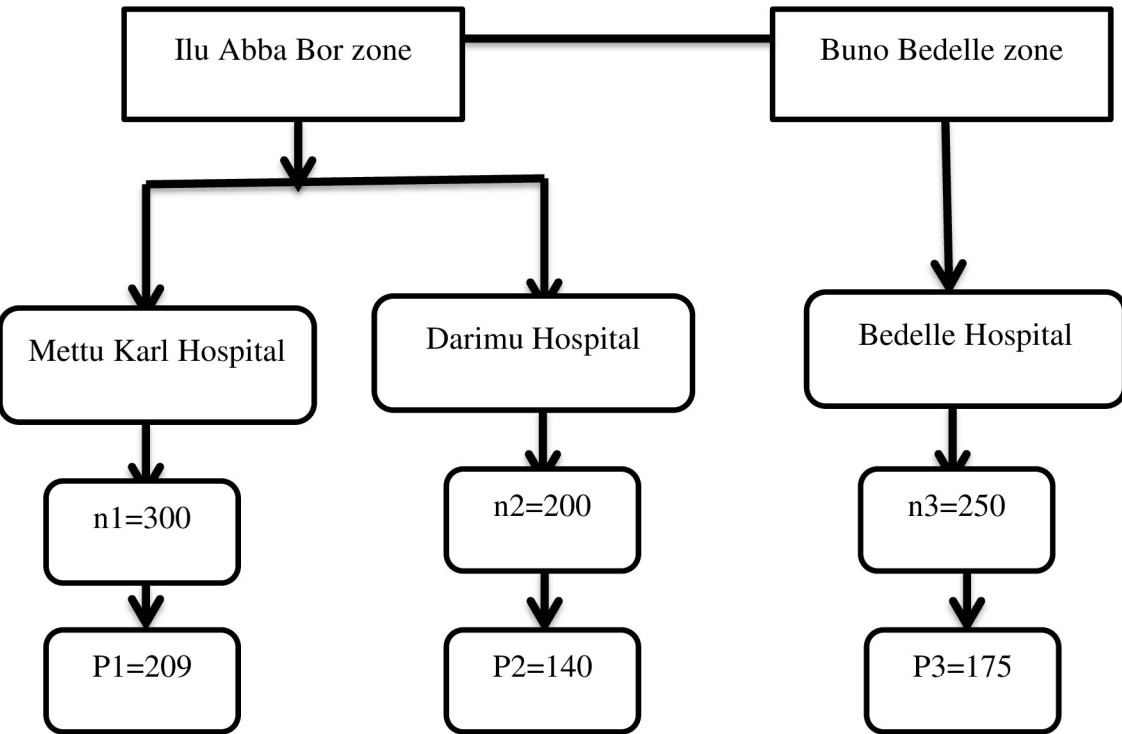

**Fig 1. The schematic presentation of the sampling procedure that was employed to select study participants from three zones, southwest, Ethiopia, 2019.** Where, n–is the average number of schizophrenic patients who were treated at the psychiatric clinic of each hospital per one month (data collection period) as reviewed from the patients' registration book. P–is the number of schizophrenic patients who are allocated proportionally to the hospitals.

diagnostic criteria of the Diagnostic Statistical Manual of Mental Disorders, 4[th] and or 5[th] edition (DSM-IV/DSM-V) and currently attending their treatments at health facilities (hospitals). The diagnosis of the patients was first confirmed by reviewing patients' cards prior to starting data collection (interviewing the patients) and also interviewer's perception of the patient capacity was determined based on the patient level of remission. Fagerstrom Test for Nicotine Dependence (FTND) has six items, with a total score ranging from 0 to 10 was used to measure nicotine dependence [21]. The FTND has been shown to have good test-retest reliability and validity in populations of smokers with mental health problems [22]. At a cut-off score ≥of 5, the FTND has good sensitivity and specificity (0.75 and 0.80, respectively) [23] and was considered to indicate tobacco dependence.

Alcohol use disorders (AUDs) were assessed using the four-item CAGE questionnaire (Cut down, Annoyed, Guilty, and Eye-opener). CAGE is short and easily applied in clinical practice. The sensitivity and specificity of CAGE at a cut-off score $\geq 2$ was 0.71 and 0.90, respectively [11]. In this study, a total score ≥2 on CAGE was used to indicate an alcohol use disorder.

The severity of Dependence Scale (SDS) was used to assess Khat use disorder. It is a screening tool for the Diagnostic and Statistical Manual of Mental Disorders, Fifth Edition (DSM-5) - defined Khat use disorder [24]. SDS is a brief and simple screening tool that was validated in Mizan, the Southwestern part of Ethiopia to identify individuals experiencing a Khat use disorder syndrome and experiencing high rates of adverse consequences in association with the use [25]. Each of the five items is scored on a 4-point scale (0–3). The total score is obtained through the addition of the 5-item ratings.

## Data processing and analysis

Epi Data Version 3.1 was used for data entry following checks and coded for. Then, the data were exported to the Statistical Package for Social Science Version 24.0 for further analysis. Simple descriptive statistics (median, percentage, frequencies, and interquartile range) were used to compute demographic characteristics of participants. In addition, bivariable analysis was used to see the significance of the association. Variables that showed strong association (p-value <0.25) in bivariate analysis were entered into multivariable logistic regressions to identify independently associated variables. Multicollinearity was checked by the Variance Inflation Factor (VIF). Statistical significance was declared at a p-value less than 0.05. The significance of the association of the variables was described using Adjusted Odds Ratio (AOR) with a 95% confidence interval.

## Data quality control

The questionnaire was prepared first in English and translated into Afaan Oromo/Amharic then back-translated to English by another person who was blinded for the English version to check the clarity of the questionnaire. To identify potential problems and to make important modifications, the questionnaire was pre-tested on 5% of the total study participants were randomly selected in the same population outside the study area in Jimma Medical center psychiatric clinic one week before the actual data collection date. The prepared questionnaire was checked thoroughly for its completeness, objective, and variable before it was distributed to respondents. Also, the collected data were checked for its completeness. The supervisor was three first-degree holder instructors. A pre-test was done after training is given to the supervisors on how to supervise data collection. The principal investigator checked for the completeness of filling questionnaires at the end of each data collection date. Any error, ambiguity, incompleteness, or another encountered problem was addressed immediately after the supervisor receives the filled questionnaire from each data collector.

## Operational definition

**Tobacco dependence** = individuals who score FTND 5 and above.

- A total FTND score of five indicates moderate nicotine dependence,

- A a score of 6–7 indicates high nicotine dependence and

- A a score of 8–10 indicates very high nicotine dependence

   **Schizophrenia:** is a clinical diagnosis reached by a clinician based on DSM-IV/DSM-5 diagnostic criteria as reviewed from the patient card.
   **Physical illness:** is any diagnosed medical problem like hypertension, diabetes Mellitus, heart failure made by the clinician during the follow-up period.
   **Substance use:** ever use of any psychoactive substance in the past 12 months.

## Ethical clearance

Ethical clearance was obtained from the Research, Ethical Review Board of Mettu University, college of health sciences, and the study was done according to the declaration of Helsinki. And also an approval letter was obtained from the head department of psychiatry. After the ethical review board has approved the consent procedure, selected participants were told about the nature, purposes, benefits, and adverse effects of the study and invited to participate. Participants were told the right to refuse or discontinue participation at any time they want. Confidentiality was ensured and all related questions, they raised were answered during data collection. Written informed consent was obtained from study participants.

## Results

### Socio-demographic characteristics of the study participants

A total of 524 participants was participating in the study, of which 515 responded, giving a response rate of 98.3%. The mean age (±SD) of the study, participants were 33.7 (±7.9) years of age. About 287 (55.7%) of the respondents were married. Among the respondents, 341 (66.2%) were male, 326 (63.3%) were Oromo, and 150 (29.1%) of the study participants had attended primarily (grade 1–8) education More than half of them had a family size of four or above and the median monthly incomes of the respondents were 700ETB, which ranges from 100-5000ETB. (**Table 1**)

   **Clinical and other substance-related characteristics of patients with schizophrenia.** More than half of the participants were attending their treatment for less than 6years (56.7%). Of patients with a history of admission, about 17.1% were admitted 2times for their condition. Few of them had both family history of mental illness (15.7%) and substance use (17.1%). About one third (36.3%) of the study participants fulfilled alcohol use disorder using CAGE criteria. (**Table 2**)

### Prevalence of tobacco dependence among patients with schizophrenia

The prevalence of tobacco dependence among patients with schizophrenia was 22.3% 95% CI (18.6, 26). Concerning the severity of tobacco dependence, about 3.5%, 13.8% and 5% of the respondents use moderate, high, and very excessive levels of tobacco dependence respectively. (**Table 3**)

   More than half of the respondents, 308 (59.8), smoked less than 10 items of cigarettes on a daily basis. About one-fourth of schizophrenic patients with tobacco dependence smoke cigarettes within 5-30miutes soon after waking. The proportions of tobacco dependence among male schizophrenic patients 88 (25.8%) were higher compared to their counterparts 27 (15.5%).

**Table 1. Socio-demographic characteristic of patients with schizzophrenias at southwest Ethiopia, 2019 (n = 515).**

| Variables | Category | Numbers (n) | Percentage (%) |
|---|---|---|---|
| Sex | Male | 341 | 66.2 |
| | Female | 174 | 33.8 |
| Age(in years) | 18–24 | 59 | 11.5 |
| | 25–34 | 215 | 41.7 |
| | 35–44 | 178 | 34.6 |
| | 45–54 | 63 | 12.2 |
| Religion | Muslim | 332 | 64.5 |
| | Orthodox | 101 | 19.6 |
| | Protestant | 82 | 15.9 |
| Ethnicity | Oromo | 326 | 63.3 |
| | Amhara | 116 | 22.5 |
| | SNNP | 73 | 14.2 |
| Marital status | Single | 175 | 34.0 |
| | Married | 287 | 55.7 |
| | Divorced/widowed | 53 | 10.3 |
| Educational status | No formal education | 132 | 25.6 |
| | Primarily school | 150 | 29.1 |
| | Secondary | 116 | 22.5 |
| | Above secondary | 117 | 22.7 |
| Residence | Urban | 292 | 56.7 |
| | Rural | 223 | 43.3 |
| Occupation | Government employed | 103 | 20.0 |
| | Self-employed | 185 | 35.9 |
| | Unemployed | 227 | 44.1 |
| Family size | <4 | 211 | 41.0 |
| | ≥4 | 304 | 59.0 |
| Monthly Income | <700 ‖ ETB | 264 | 51.3 |
| | ≥700 ETB | 251 | 48.7 |

SNNP (South nations and nationalities and peoples)–stands for Kaffa, Dawuro, Yem, Walayta, Gurage and Silte, Median of monthly income, ETB–Ethiopian birr.

### Correlates of tobacco dependence among patients with schizophrenia

In the univariable logistic regression, different factors have been shown to have associated with tobacco dependence among patients with schizophrenia. Accordingly, male gender, unemployment, being on treatment for 5years, having a history of admission and frequent admission, presence of physical illness, family history of mental illness, being educated above secondary school

After controlling for cofounders, male gender (AOR 2.19, 95% CI = 1.25, 3.83), being on treatment for more than 5years (AOR 4.37, 95% CI = 2.11, 9.02), having a history of admission (AOR 4.01, 95% CI = 1.99, 8.11), and family history of mental illness (AOR 1.90, 95% CI = 1.04, 3.48) were shown to have a significant positive association with tobacco dependence in the final regression analysis. (**Table 4**)

## Discussion

A cross-sectional study was conducted in three hospitals located in the southwestern part of Ethiopia revealed about one-quarter of patients with schizophrenia reported tobacco

**Table 2. Clinical and other substance-related characteristics of patients with schizophrenia, southwest Ethiopia, 2019 (n = 515).**

| Variables | Category | Numbers (n) | Percentage (%) |
|---|---|---|---|
| Admission history | No | 308 | 59.8 |
| | Yes | 207 | 40.2 |
| Frequency of admission | 1 | 75 | 14.6 |
| | 2 | 88 | 17.1 |
| | 3 | 44 | 8.5 |
| Physical illness | Yes | 38 | 7.4 |
| | No | 477 | 92.6 |
| Family history of mental illness | No | 434 | 84.3 |
| | Yes | 81 | 15.7 |
| Family history of substance use | No | 427 | 82.9 |
| | Yes | 88 | 17.1 |
| Alcohol use disorder | Yes | 187 | 36.3 |
| | No | 328 | 63.7 |
| Khat dependence | Yes | 155 | 30.1 |
| | No | 360 | 69.9 |
| Duration of illness | <6 years | 292 | 56.7 |
| | ≥6 years | 223 | 43.3 |
| Treatment duration | <5 years | 271 | 52.6 |
| | ≥5 year s | 244 | 47.7 |

dependence. As the majority of the previously conducted studies targeted cigarette smoking among patients with schizophrenia, however, the current study aimed to determine the prevalence of tobacco dependence and its correlates among patients with schizophrenia. So, this could have additional benefits set direction or develop strategies to deal with the impacts of the problem.

The overall prevalence of tobacco dependence among schizophrenic patients was 25.9%. These results were higher compared to the prevalence of tobacco dependence among the general population in Ethiopia which is 7.9% [26]. This difference could be due to the chronic nature of the illness and is used as a form of self-medication, normalizing some central nervous system deficits involved in the disorder. The results were also higher compared to the finding of the study from Nigeria (20.4%) [19]. The possible difference might be explained due to differences in study instruments (FTND vs. PSE-10). However, the finding of the current study was lower than the result of a study from Turkey 49% [18], United Kingdom 44.6% [13], India

**Table 3. Level of tobacco dependence and frequency of smoking amongst patients with schizophrenia attending their treatments at southwest Ethiopia, 2019 (n = 515).**

| Variables | | Numbers (n) | Percentage (%) |
|---|---|---|---|
| Level of dependence | Moderate (5) | 18 | 3.5 |
| | High (6–7) | 71 | 13.8 |
| | Very high (8–10) | 26 | 5 |
| Frequency of smoking | Never | 401 | 77.9 |
| | Once or twice | 34 | 6.6 |
| | Daily or almost daily | 37 | 7.2 |
| | Weekly | 21 | 4.1 |
| | Monthly | 22 | 4.3 |

**Table 4. Factors associated with tobacco dependence among schizophrenic patients at Mettu Karl referral, Bedelle and Agaro hospitals, southwest Ethiopia, 2020 (N = 515).**

| Variable | Category | Tobacco use | | COR, 95% (CI) | AOR, 95% (CI) |
|---|---|---|---|---|---|
| | | Not dependent N (%) | Dependent N (%) | | |
| Sex | Female | 147 (84.5) | 27 (15.5) | Ref | Ref |
| | Male | 253 (74.2) | 88 (25.8) | 1.89 (1.18,3.05) | **2.19 (1.25,3.83)**** |
| Occupational status | Government employed | 79 (76.9) | 24 (23.3) | Ref | Ref |
| | Self-employed | 168 (90.8) | 17 (9.2) | 0.33 (0.17,0.66) | 0.41 (0.17, 1.00) |
| | Unemployed | 153 (67.4) | 74 (32.6) | 1.59 (0.93,2.72) | 2.09 (0.98, 4.49) |
| Frequency of admission | None | 146 (71.9) | 57 (28.1) | Ref | Ref |
| | 1 | 124 (80.0) | 31 (20.0) | 0.79 (0.43,1.45) | 1.32 (0.66, 2.65) |
| | 2 | 92 (81.4) | 21 (18.6) | 0.55 (0.29,1.03) | 1.63 (0.66,4.01) |
| | 3 | 38 (86.4) | 6 (13.6) | 0.46 (0.19,1.12) | 0.67 (0.23,2.15) |
| Duration of treatment | <5years | 220 (81.2) | 51 (18.8) | Ref | Ref |
| | ≥5years | 180 (73.8) | 64 (26.2) | 1.53 (1.01,2.33) | **4.37 (2.11,9.02)***** |
| Educational status | No formal education | 112 (84.8) | 20 (15.2) | Ref | Ref |
| | Primarily school | 117 (78.0) | 33 (22.0) | 1.58 (0.86, 2.92) | 0.59 (0.25, 1.44) |
| | Secondary | 88 (75.9) | 28 (24.1) | 1.78 (0.94,3.37) | 0.91 (0.41,2.00) |
| | Above | 83 (70.9) | 34 (29.1) | 2.29 (1.23,4.27) | 0.73 (0.34,1.59) |
| Admission history | No | 265 (86.0) | 43 (14.0) | Ref | Ref |
| | Yes | 135 (65.2) | 72 (34.8) | 3.29 (2.14,5.06) | **4.01 (1.99, 8.11)**** |
| Physical illness | Yes | 25 (65.8) | 13 (34.2) | 1.91 (0.95,3.87) | 1.79 (0.73,4.43) |
| | No | 375 (78.6) | 102 (21.4) | Ref | Ref |
| Alcohol use disorder | Dependent | 151 (80.7) | 36 (19.3) | 1.33 (0.86,2.07) | 1.12 (0.67,1.85) |
| | Non-dependent | 249 (75.9) | 79 (24.1) | Ref | Ref |
| Family history of mental illness | No | 347 (80.0) | 87 (20.0) | Ref | Ref |
| | Yes | 53 (65.4) | 28 (34.6) | 2.11 (1.26,3.53) | **1.90 (1.04,3.48)*** |

Abbreviations: OR, Odds Ratio; CI, Confidence Interval. Ref: Reference category NB.*Persisted significant at P-value <0.05, ** significant at P-value ≤0.001. *** Significant at P-value <0.0001.

68.8% [27] and Ethiopia 29.1% [20]. The discrepancy might be attributable to the difference in study design, instruments, and study setting.

After controlling for confounders, the odds of developing tobacco dependence among patients who attended treatment for more than 5 years were 4.4 times higher than those who attended for less than 5years. This was in agreement with the findings of the study conducted in China [28] and the USA [29]. It is clear that, from the nature of Schizophrenia, at the time of treatment progress or illness become episodic, the intensity increase over time, exposing them to use tobacco in the form of self-medication [30]. Generally, the longer the duration of the treatment period, the higher the danger of developing tobacco dependence.

Individual patients having a family history of mental illness were 1.9times more likely to develop tobacco dependence compared to those without a family history of mental illness which was supported by the finding of previously conducted studies [31]. The notion was indicating the role of genetic factors in the etiology of smoking behaviour and the high comorbidity between nicotine dependence and schizophrenia [32].

Furthermore, gender was shown a significant association with tobacco dependence in patients living with schizophrenia was gender. Accordingly, in the current study, males were 2times more likely to develop tobacco dependence compared to their counterparts and this was in agreement with the finding of the previous study [14, 33]. The possible reason might be

related to the fact that male schizophrenic patients were more likely to smoke tobacco as they experience a lesser intensity of negative symptoms compared to females [34].

Finally, the current study revealed patients with schizophrenia who was a history of admission were more likely to develop tobacco dependence compared to those attending their treatment on an out-patient basis. This was in line with the findings of the previously published studies [14]. The finding of the study supports an association of illness severity with admission history [35].

## Limitations

One of the limitations of this study might be the cross-sectional study design, which does not allow causal inference. Again In the current study, patients who were living with schizophrenia and schizophrenia-like disorder attending inpatient care were excluded. Thus, the finding of the study may not be generalizable to all patients with schizophrenia or like disorder. Also, the lack of any scale measuring current psychopathology was another limitation of this study.

## Conclusions

A study shows a significant the proportion of tobacco dependence among people living with schizophrenia. Factors like, being male gender, being on treatment for more than 5 years, having a history of admission, and family history of mental illness was found to have a significant positive association with tobacco dependence. Hence, there is a need for coordinated and comprehensive management clinically to manage tobacco dependence along with identified risk factors in patients with schizophrenia. Also the finding call for the clinicians, managers, ministry of health and other stakeholders on the substance use prevention strategies that target personal and environmental control.

## Acknowledgments

We would like to thank Mettu University for granting ethical approval and funding the study. Our deepest thanks go to all study participants, data collectors, and supervisors who spent their valuable time for the good outcome of the research work.

## Author Contributions

**Conceptualization:** Defaru Desalegn, Zakir Abdu, Mohammedamin Hajure.

**Data curation:** Defaru Desalegn, Zakir Abdu, Mohammedamin Hajure.

**Formal analysis:** Defaru Desalegn, Zakir Abdu, Mohammedamin Hajure.

**Funding acquisition:** Defaru Desalegn, Zakir Abdu, Mohammedamin Hajure.

**Investigation:** Defaru Desalegn, Zakir Abdu, Mohammedamin Hajure.

**Methodology:** Defaru Desalegn, Zakir Abdu, Mohammedamin Hajure.

**Supervision:** Defaru Desalegn, Zakir Abdu, Mohammedamin Hajure.

**Validation:** Zakir Abdu, Mohammedamin Hajure.

**Visualization:** Defaru Desalegn, Zakir Abdu, Mohammedamin Hajure.

**Writing – review & editing:** Defaru Desalegn, Zakir Abdu, Mohammedamin Hajure.

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
