## [Decision Letter · Decision Letter 0]

15 Jan 2021

PONE-D-20-31418

Prevalence of Tobacco Dependence and Associated Factors among Patients with Schizophrenia Attending Their Treatments at Southwest Ethiopia; Hospital-Based Cross-Sectional Study

PLOS ONE

Dear Dr. Desalegn,

Thank you for submitting your manuscript to PLOS ONE. After careful consideration, we feel that it has merit but does not fully meet PLOS ONE’s publication criteria as it currently stands. Therefore, we invite you to submit a revised version of the manuscript that addresses the points raised during the review process.

We look forward to receiving your revised manuscript.

Kind regards,

Stanton A. Glantz

Academic Editor

PLOS ONE

2. Please describe in your methods section how capacity to provide consent was determined for the participants in this study. Please also state whether your ethics committee or IRB approved this consent procedure. If you did not assess capacity to consent please briefly outline why this was not necessary in this case.

3. In your Methods section, please provide additional information about the participant recruitment method and the demographic details of your participants. Please ensure you have provided sufficient details to replicate the analyses such as: a) the recruitment date range (month and year), b) a description of how participants were recruited, and c) descriptions of the specific locations where participants were recruited and where the research took place.

4. Please include additional information regarding the interviewer-administered structured questionnaire used in the study and ensure that you have provided sufficient details that others could replicate the analyses. For instance, if you developed a questionnaire as part of this study and it is not under a copyright more restrictive than CC-BY, please include a copy, in both the original language and English, as Supporting Information.

7. Thank you for submitting the above manuscript to PLOS ONE. During our internal evaluation of the manuscript, we found significant text overlap between your submission and the following previously published works.

- https://moam.info/abstract-introduction-clinical-neuropsychiatry_5a856eba1723dd57bb3c06ad.html

- https://www.hindawi.com/journals/jad/2018/8102165/

- https://www.mdpi.com/1660-4601/10/10/4790/htm

- https://worldwidescience.org/topicpages/r/receiving+inpatient+psychiatric.html

- https://www.dovepress.com/suicidal-behavior-and-associated-factors-among-students-in-mettu-unive-peer-reviewed-fulltext-article-PRBM

Please revise the manuscript to rephrase the duplicated text, cite your sources, and provide details as to how the current manuscript advances on previous work. Please note that further consideration is dependent on the submission of a manuscript that addresses these concerns about the overlap in text with published work.

Reviewers' comments:

Reviewer's Responses to Questions

**Comments to the Author**

1. Is the manuscript technically sound, and do the data support the conclusions?

Reviewer #1: Partly

Reviewer #2: No

2. Has the statistical analysis been performed appropriately and rigorously? 

Reviewer #1: No

Reviewer #2: Yes

3. Have the authors made all data underlying the findings in their manuscript fully available?

Reviewer #1: No

Reviewer #2: Yes

4. Is the manuscript presented in an intelligible fashion and written in standard English?

Reviewer #1: Yes

Reviewer #2: No

5. Review Comments to the Author

Reviewer #1: Abstract

In the first sentence the word ‘abused’ and ‘abuser’ is not right. There is no diagnostic criteria for tobacco abuse in DSM-5.

The introduction part can be shortened. There are a lot of research findings from all the over the world. It should be summarized to the most relevant researches and high quality evidence.

Materials and methods

Please include a statement on the type of the study design.

Please include some statements about the interviewer’s level of expertise. And the instrument that has been used to assess tobacco dependence (Fagerstrom Test for Nicotine Dependence (FTND)), was the instrument used in it’s original language or was it translated?

If it was translated, please specify the process of translation of this or other instruments.

Please put the reference paper which has validated the SDS instrument in Ethiopia.

Result

The 1st paragraph second sentence include the ‘mean age’. On the next sentences there are repetitions, please correct it. The last sentence the median (IQR) monthly income the IQR is not specified in a range of numbers.

2nd paragraph 3rd line include ‘family history’

Correlates of tobacco dependence among patients with schizophrenia

On the bivariate logistic regression,-- it is univariable logistic regression

There is inconsistency of the result from the table 3 and the above paragraph presented about the significant result on univariable and multivariable logistic regression.

Discussion

It needs further explanation on recommendation and the uniqueness of this study.

Reviewer #2: This study addressed the frequency of ND in patents with schizophrenia in Ethiopia. While similar study has already been published (Molla et al, 2017), the question is whether this study brings new data. This text needs major revision, and English language revision.

Introduction

Page 9-The impact of smoking among schizophrenic patients, not only increases metabolism and vascular risks-what is meant by „metabolism “?

It decreases the antipsychotic therapeutic effects as smoking induces the medication metabolism in the liver reducing up to 48% the active metabolites in serum (9). -please, be more specific-because smoking does not induce the metabolism of all antipsychotics

One hospital based cross sectional study done among 429 inpatients schizophrenic patients in China receive antipsychotic mono-therapy found that the prevalence rate of current smoking was 40.6%and 57.5% in males and 6.3% in females. -please, put the reference number in the parenthesis, and correct English grammar

A cross sectional study done in Singapore among male schizophrenic patients found that the lifetime prevalence and current smoker are 54.1%and 42.4%respectively (15). -life time prevalence of what?

Methods

How was schizophrenia confirmed? According to which classification system? It is unclear whether the patients were in-or out-patients or both.

Exclusion criteria-„patients who are seriously ill “-please define what this term refers to

Operational definitions: Chronic illness: past mental illness...how could past mental illnesses be defined, when participants already have schizophrenia?

The study lacks any measure of schizophrenia psychopathology (such as PANSS)

What about pharmacotherapy? Did patients receive antipsychotics?

Results

The mean (±SD) of the study participants were 33.7(±7.9) years-the term „age “is missing

66.2% were males-was mentioned twice

The section: „ Prevalence of tobacco dependence among patients with schizophrenia “is unclear and difficult to read. How many patients were current smokers? The data providing number (frequency) of smokers, and categories of FTND-defined nicotine dependence, need to be presented in a separate table

„Few of them had history of mental illness (15.7%)-it is unclear,

because all patients had schizophrenia

Discussion

Please, provide in the discussion the rates of nicotine dependence in Ethiopia general population, and then comment on the ND rate in schizophrenia patients, whether and how it differs compared to general population in the same country.

Limitations

The lack of any scale measuring current psychopathology is also a limitation.

The conclusion „a significant proportion of tobacco dependence...“ would be valid only if this ND frequency outnumbers smoking prevalence in Ethiopia general population.

6. PLOS authors have the option to publish the peer review history of their article (what does this mean?). If published, this will include your full peer review and any attached files.

Reviewer #1: No

Reviewer #2: No

---

## [Author Response · Author response to Decision Letter 0]

26 Jun 2021

Dear PLOS ONE,

Thank you for your comments, concerns and consideration of our manuscript. We tried to incorporate for comments and answer for comments written by yellow color 

1. Please ensure that your manuscript meets PLOS ONE's style requirements, including those for file naming

Authors’ response: we did accept the comment and tried to address as our manuscript meets

2. Please describe in your methods section how capacity to provide consent was determined for the participants in this study. Please also state whether your ethics committee or IRB approved this consent procedure. If you did not assess capacity to consent please briefly outline why this was not necessary in this case.

Authors’ response: we have descried how capacity to provide consent and also stated as our ethical committee approved this consent procedure. Please see under ‘ethical clearance’ subtitle of materials ad methods section

3. In your Methods section, please provide additional information about the participant recruitment method and the demographic details of your participants. Please ensure you have provided sufficient details to replicate the analyses such as: 

a) the recruitment date range (month and year),

 b) a description of how participants were recruited, and 

c) descriptions of the specific locations where participants were recruited and where the research took place.

Authors’ response: we have provided sufficient additional information about the participant recruitment method and the demographic details of our participants. Please see under materials and methods section 

4. Please include additional information regarding the interviewer-administered structured questionnaire used in the study and ensure that you have provided sufficient details that others could replicate the analyses. For instance, if you developed a questionnaire as part of this study and it is not under a copyright more restrictive than CC-BY, please include a copy, in both the original language and English, as Supporting Information.

Authors’ response: we have used the interviewer administered questionnaire to collect data in patients with schizophrenia since they don’t have similar level of educational background to clearly understand the queries and to reduce information bias.

5. We note that you have indicated that data from this study are available upon request. PLOS only allows data to be available upon request if there are legal or ethical restrictions on sharing data publicly.

Authors’ response: All relevant data are included within the paper. The data would be guarded carefully by our research team for the only purpose of this scientific study and it is an ongoing project. Participants were not signed consent for data publicity. For all these reasons and following the indicators of the research review committee of college of health sciences, Mettu University, the authors must not upload the dataset to a stable, public repository. Interested, qualified researchers can access the data by requesting Dean College of health sciences of Mettu University, Desalegn Chilo (desalegchilo89@gmail.com) and the corresponding author, Defaru Desalegn (defdesalegn2007@gmail.com) 

Authors’ response: we have deleted from any other section and we stated only in methods section

7. Thank you for submitting the above manuscript to PLOS ONE. During our internal evaluation of the manuscript, we found significant text overlap between your submission and the following previously published works.

Please revise the manuscript to rephrase the duplicated text, cite your sources, and provide details as to how the current manuscript advances on previous work. Please note that further consideration is dependent on the submission of a manuscript that addresses these concerns about the overlap in text with published work.

Authors’ response: we accept the comments and rephrased the documents for duplicated contents.

Reviewers' comments:

Reviewer's Responses to Questions

Comments to the Author

1. Is the manuscript technically sound, and do the data support the conclusions?

Reviewer #1: Partly

Reviewer #2: No

 Authors’ response: we accepted the comments and we have re-write our conclusion 

2. Has the statistical analysis been performed appropriately and rigorously? 

Reviewer #1: No

Reviewer #2: Yes

 Authors’ response: Yes, we have used the appropriate statistical analysis …first after data were collected we checked, coded and entered into Epi data Version 3.1. Then, we exported to SPSS Version 24.0 for analysis. Assumptions were checked and bivariate and multivariate logistic analysis were done…

3. Have the authors made all data underlying the findings in their manuscript fully available?

Reviewer #1: No

Reviewer #2: Yes

Authors’ response: All relevant data are included within the paper. The data would be guarded carefully by our research team for the only purpose of this scientific study and it is an ongoing project. Participants were not signed consent for data publicity. For all these reasons and following the indicators of the research review committee of college of health sciences, Mettu University, the authors must not upload the dataset to a stable, public repository. Interested, qualified researchers can access the data by requesting Dean College of health sciences of Mettu University, Desalegn Chilo (desalegchilo89@gmail.com) and the corresponding author, Defaru Desalegn (defdesalegn2007@gmail.com) 

4. Is the manuscript presented in an intelligible fashion and written in Standard English?

Reviewer #1: Yes

Reviewer #2: No

Authors’ response: we accepted the comment and thoroughly edited the whole document for grammatical error or any other unclear contents.

5. Review Comments to the Author

Reviewer #1: Abstract

In the first sentence the word ‘abused’ and ‘abuser’ is not right. There is no diagnostic criteria for tobacco abuse in DSM-5.

Authors’ response: Accepted and replaced the word ‘abused’ and ‘abuser’ with the word ‘used’ and ‘user’ respectively. 

The introduction part can be shortened. There are a lot of research findings from all the over the world. It should be summarized to the most relevant researches and high quality evidence.

Authors’ response: we accepted the comments and we have shortened the introduction.

Materials and methods

Please include a statement on the type of the study design.

Authors’ response: Study design was included in the manuscript

Please include some statements about the interviewer’s level of expertise.

Authors’ response: we have included the statements about the interviewer’s level of expertise under ‘data collection procedures and tools’ subtitle of methods section.

 And the instrument that has been used to assess tobacco dependence (Fagerstrom Test for Nicotine Dependence (FTND)), was the instrument used in it’s original language or was it translated?

Authors’ response: we used the translated instrument

If it was translated, please specify the process of translation of this or other instruments.

Authors’ response: the process of translation was described under ‘data quality control’ subtitle of methods section 

Please put the reference paper which has validated the SDS instrument in Ethiopia.

Authors’ response: Can be added, because it was validated at Mizan, Ethiopia

Result

The 1st paragraph second sentence include the ‘mean age’. 

Authors’ response: we have included 

On the next sentences there are repetitions, please correct it. 

Authors’ response: we have omitted the repeated sentences 

The last sentence the median (IQR) monthly income the IQR is not specified in a range of numbers. 

Authors’ response: we accept the comment and corrected accordingly ‘‘the median monthly income of the respondents were 700ETB, which ranges from 100-5000ETB and the interquartile range is 1000.’’

2nd paragraph 3rd line include ‘family history’

Authors’ response: we have Included 

Correlates of tobacco dependence among patients with schizophrenia

On the bivariate logistic regression,-- it is univariable logistic regression

There is inconsistency of the result from the table 3 and the above paragraph presented about the significant result on univariable and multivariable logistic regression.

Authors’ response: In our analysis part, for univariable analysis we have used a P –value of 0.25 or less as inclusion criteria for the final model so as not to miss important clinical variables. Accordingly male gender, unemployment, being on treatment for 5years, having a history of admission and frequent admission, presence of physical illness, family history of mental illness and being educated above secondary school. However, for the gender, we mistakenly stated and admit to edit to male.

Discussion

It needs further explanation on recommendation and the uniqueness of this study.

Authors’ response: we accept and addressed it

Reviewer #2: This study addressed the frequency of ND in patents with schizophrenia in Ethiopia. While similar study has already been published (Molla et al, 2017), the question is whether this study brings new data. This text needs major revision, and English language revision.

Authors’ response: our study addressed specifically the frequency of ND in patients with schizophrenia in Ethiopia. But, this (Molla et al, 2017) study, the previously done and published, was done among mental illness in general. Also our study assessed factors like Khat and alcohol by using independent instruments. In addition, we have revised the whole document for grammar and spelling error.

Introduction

Page 9-The impact of smoking among schizophrenic patients, not only increases metabolism and vascular risks-what is meant by „metabolism “?

Authors’ response: It mean that smoking increases the activity of cytochrome p450 isoenzyme 1A2 (CYP1A2) and UDP-glucuronosyltransefereses (UGT), which are responsible for drug metabolism (antipsychotics)

It decreases the antipsychotic therapeutic effects as smoking induces the medication metabolism in the liver reducing up to 48% the active metabolites in serum (9). -please, be more specific-because smoking does not induce the metabolism of all antipsychotics

Authors’ response: we have addressed above 

One hospital based cross sectional study done among 429 inpatients schizophrenic patients in China receive antipsychotic mono-therapy found that the prevalence rate of current smoking was 40.6%and 57.5% in males and 6.3% in females. -please, put the reference number in the parenthesis, and correct English grammar

Authors’ response: we accepted the comment and corrected accordingly

A cross sectional study done in Singapore among male schizophrenic patients found that the lifetime prevalence and current smoker are 54.1%and 42.4%respectively (15). -life time prevalence of what?

Authors’ response: we have corrected as “life time prevalence of smoking cigarette …”

Methods

How was schizophrenia confirmed? According to which classification system? It is unclear whether the patients were in-or out-patients or both.

Authors’ response: as we described under operation definition schizophrenia is a clinical diagnosis reached by clinician based on DSM-IV or DSM-5 diagnostic criteria as reviewed from patient card and the study populations were sample of patients with schizophrenia who attended the out-patient treatment. Please under ‘study population’ subtitle of methods section

Exclusion criteria-„patients who are seriously ill “-please define what this term refers to

Authors’ response: we have defined it. Please see under ‘inclusion and exclusion criteria’ subtitle of methods section

Operational definitions: Chronic illness: past mental illness...how could past mental illnesses be defined, when participants already have schizophrenia?

Authors’ response: We accept the comments and corrected

The study lacks any measure of schizophrenia psychopathology (such as PANSS)

Authors’ response: yes, we didn’t assessed psychopathology and we accepted as the limitation of our study 

What about pharmacotherapy? Did patients receive antipsychotics?

Authors’ response: Yes, but we didn’t assessed the type of antipsychotic they are taking

Results

The mean (±SD) of the study participants were 33.7(±7.9) years-the term „age “is missing

66.2% were males-was mentioned twice

Authors’ response: we accepted the comments and corrected

The section: „ Prevalence of tobacco dependence among patients with schizophrenia “is unclear and difficult to read. How many patients were current smokers? The data providing number (frequency) of smokers, and categories of FTND-defined nicotine dependence, need to be presented in a separate table

Authors’ response: we accept the comment and incorporated the points in result part in Table 3. for the rate of current , the tools will not assess in terms of current smokers rather on daily ,weekly and monthly basis as stated in Table 3.

„Few of them had history of mental illness (15.7%)-it is unclear,

because all patients had schizophrenia

Authors’ response: we appreciate your concern and corrected below. Indeed we mean to family history of mental illness rather than personal history including schizophrenia and other diagnosable mental illness. ‘‘Few of them had both family history of mental illness (15.7%) and substance use (17.1%)’’. 

Discussion

Please, provide in the discussion the rates of nicotine dependence in Ethiopia general population, and then comment on the ND rate in schizophrenia patients, whether and how it differs compared to general population in the same country.

Authors’ response: we accepted the comments and we have provided the rates of nicotine dependence in Ethiopia general population in discussion and have compared with the current findings in schizophrenic patients.

Limitations

The lack of any scale measuring current psychopathology is also a limitation.

Authors’ response: we accepted the comments and added on the limitation part

The conclusion „a significant proportion of tobacco dependence...“ would be valid only if this ND frequency outnumbers smoking prevalence in Ethiopia general population.

6. PLOS authors have the option to publish the peer review history of their article (what does this mean?). If published, this will include your full peer review and any attached files.

Do you want your identity to be public for this peer review? For information about this choice, including consent withdrawal, please see our Privacy Policy.

Reviewer #1: No

Reviewer #2: No

 Authors’ response: No 

While revising your submission, please upload your figure files to the Preflight Analysis and Conversion Engine (PACE) digital diagnostic tool, https://pacev2.apexcovantage.com/. PACE helps ensure that figures meet PLOS requirements. To use PACE, you must first register as a user. Registration is free. Then, login and navigate to the UPLOAD tab, where you will find detailed instructions on how to use the tool. If you encounter any issues or have any questions when using PACE, please email PLOS at figures@plos.org. Please note that Supporting Information files do not need this step

Additional comments 

1. You state that "All relevant data are included within the paper". However, the rest of your response indicates that the relevant data are available upon request. Please confirm that your data is available upon request

Authors’ response: we admit our errors that we unintentionally responded as relevant data are available upon request and we all the authors agreed that all relevant data are included within the paper. The data would be guarded carefully by our research team for the only purpose of this scientific study and it is an ongoing project. Also, participants were not signed consent for data publicity. For all these reasons and following the indicators of the research review committee of college of health sciences, Mettu University, the authors must not upload the dataset to a stable, public repository. But, interested, qualified researchers can access the data by requesting Dean College of health sciences of Mettu University, Desalegn Chilo (desalegchilo89@gmail.com) and the corresponding author, Defaru Desalegn (defdesalegn2007@gmail.com) 

2. You state that "the data would be guarded carefully by [y]our research team". Please confirm that your data will indeed be available upon request to researchers who submit data access requests and meet the criteria for access to confidential data.

Authors’ response: Interested, qualified researchers can access the data by requesting Dean College of health sciences of Mettu University, Desalegn Chilo (desalegchilo89@gmail.com) and the corresponding author, Defaru Desalegn (defdesalegn2007@gmail.com) 

(2) Please describe in your methods section how capacity to provide consent was determined for the participants in this study.

Authors’ response: we have described how capacity to provide consent was determined for the participants in our study in our methods section specifically under ‘ethical clearance ‘ 

(3) We note that there is still some overlap within your Abstract and Introduction

Authors’ response: we appreciate your concerns and we have corrected accordingly. Please, see at our abstract and introduction.

---

## [Decision Letter · Decision Letter 1]

3 Aug 2021

PONE-D-20-31418R1

Prevalence of Tobacco Dependence and Associated Factors among Patients with Schizophrenia Attending Their Treatments at Southwest Ethiopia ; Hospital-Based Cross-Sectional Study

PLOS ONE

Dear Dr. Desalegn,

Thank you for submitting your manuscript to PLOS ONE. After careful consideration, we feel that it has merit but does not fully meet PLOS ONE’s publication criteria as it currently stands. Therefore, we invite you to submit a revised version of the manuscript that addresses the points raised during the review process.

There are still serious problems with the English.  It is important that you have the manuscript edited by a native English speaker before you submit a revised manuscript in addition to carefully addressing the reviewers' technical comments.

We look forward to receiving your revised manuscript.

Kind regards,

Stanton A. Glantz, PhD

Academic Editor

PLOS ONE

Reviewers' comments:

Reviewer's Responses to Questions

**Comments to the Author**

1. If the authors have adequately addressed your comments raised in a previous round of review and you feel that this manuscript is now acceptable for publication, you may indicate that here to bypass the “Comments to the Author” section, enter your conflict of interest statement in the “Confidential to Editor” section, and submit your "Accept" recommendation.

Reviewer #1: (No Response)

Reviewer #2: All comments have been addressed

2. Is the manuscript technically sound, and do the data support the conclusions?

Reviewer #1: Partly

Reviewer #2: Partly

3. Has the statistical analysis been performed appropriately and rigorously? 

Reviewer #1: No

Reviewer #2: I Don't Know

4. Have the authors made all data underlying the findings in their manuscript fully available?

Reviewer #1: Yes

Reviewer #2: Yes

5. Is the manuscript presented in an intelligible fashion and written in standard English?

Reviewer #1: No

Reviewer #2: No

6. Review Comments to the Author

Reviewer #1: Prevalence of Tobacco Dependence and Associated Factors among Patients with Schizophrenia Attending Their Treatments at Southwest Ethiopia ; Hospital-Based Cross-Sectional Study

Reviewer #1:

1. Abstract In the first sentence the word ‘abused’ and ‘abuser’ is not right. There is no criteria for tobacco abuse in DSM-5.

Authors’ response: Accepted and replaced the word ‘abused’ and ‘abuser’ with the word ‘used’ and ‘user’ respectively.

Re-reviewer response -Even though the word is changed the sentence is not written in a correct English grammar.

Result section of the abstract - Concerning the severity of tobacco dependence, 3.5%, 13.8% and 5% of the respondents report high and very high level of tobacco dependence respectively [ this sentence is not correct, misses the word moderate].

2. The introduction part can be shortened. There are a lot of research findings from all the over the world. It should be summarized to the most relevant researches and high quality evidence.

Authors’ response: we accepted the comments and we have shortened the introduction.

Re-reviewer response – accepted

3. Materials and methods Please include a statement on the type of the study design.

Authors’ response: Study design was included in the manuscript

Re-reviewers response – accepted

4. Please include some statements about the interviewer’s level of expertise.

Authors’ response: we have included the statements about the interviewer’s level of expertise under ‘data collection procedures and tools’ subtitle of methods section.

Re-reviewers response – accepted

5. And the instrument that has been used to assess tobacco dependence (Fagerstrom Test for Nicotine Dependence (FTND)), was the instrument used in it’s original language or was it translated?

Authors’ response: we used the translated instrument

Re-reviewers response – accepted

6. If it was translated, please specify the process of translation of this or other instruments.

Authors’ response: the process of translation was described under ‘data quality control’ subtitle of methods section

Re-reviewers response – accepted

7. Please put the reference paper which has validated the SDS instrument in Ethiopia.

Authors’ response: Can be added, because it was validated at Mizan, Ethiopia.

Re-reviewers response – if the validation paper was not published in peer reviewed journal, you cannot say that it was validated

8. Result The 1st paragraph second sentence include the ‘mean age’.

Authors’ response: we have included

Re-reviewers response – accepted

9. On the next sentences there are repetitions, please correct it.

Authors’ response: we have omitted the repeated sentences

Re-reviewers response – accepted

10. The last sentence the median (IQR) monthly income the IQR is not specified in a range of numbers.

Authors’ response: we accept the comment and corrected accordingly ‘‘the median monthly income of the respondents were 700ETB, which ranges from 100-5000ETB and the interquartile range is 1000.’’

Re-reviewers response – this is still not correct. IQR should be in a range

11. On the bivariate logistic regression,-- it is univariable logistic regression. There is inconsistency of the result from the table 3 and the above paragraph presented about the significant result on univariable and multivariable logistic regression.

Authors’ response: In our analysis part, for univariable analysis we have used a P –value of 0.25 or less as inclusion criteria for the final model so as not to miss important clinical variables. Accordingly male gender, unemployment, being on treatment for 5years, having a history of admission and frequent admission, presence of physical illness, family history of mental illness and being educated above secondary school. However, for the gender, we mistakenly stated and admit to edit to male.

Re-reviewers response – on the paragraph it is still not edited. Table 4 is not written in a consistent manner. For example the ref. value is sometime on the first row and sometimes on the second row. The analysis you have done is not correlation therefore you should not use the words like “correlations”.

12. Discussion It needs further explanation on recommendation and the uniqueness of this study. Authors’ response: we accept and addressed it

Re-reviewers response – not addressed.

On the 4th paragraph 3rd line it says “The notion was indicating the role of genetic factors and nicotine dependence in the pathogenesis of schizophrenia” this needs further explanation. Are you saying that nicotine dependence is causing schizophrenia?

On the conclusion part it says female gender is associated with tobacco dependence and it should male gender.

In general the whole text needs English language revision. The whole manuscript is written in mundane fashion it should be written in attractive way to the readers.

Reviewer #2: The authors have addressed all comments, but there are still some issues in the text that must be improved

Abstract, Background: „Tobacco smoking is that the most typically employed in patients with mental disorders; among them, patients with schizophrenia area unit the very best users...“ This text is not clear, please, change

Abstract, results: Concerning the severity of tobacco dependence, 3.5%, 13.8% and 5% of the respondents report high and very high level of tobacco dependence respectively-all three percentages need description., i.e., to which two of the three mentioned percentages do belong high and very high level of ND?

Results

The mean age (±SD) of the study participants were 33.7(±7.9) year age..-please, delete „age“ at the end of the sentence

Table 3. Please, explain what is „frequency“ and „percentage“. Does the „frequency“ actually refer to absolute numbers?

While male gender was associated with tobacco dependence in univariable logistic regression, female gender was associated with tobacco dependence in the final regression analysis. It is difficult to understand, so, please, explain this finding, and mention this also in the discussion.

Discussion

„The results were also higher compared to the finding of the study from Nigeria (20.4)“-percent?

References

References are not complete. Please, check and correct carefully all references.

For example, reference No 10 is lacking five additional authors, and the journal name. References 11 and 12 look very much alike, while the No 12 has no journal name

7. PLOS authors have the option to publish the peer review history of their article (what does this mean?). If published, this will include your full peer review and any attached files.

Reviewer #1: No

Reviewer #2: No

---

## [Author Response · Author response to Decision Letter 1]

25 Sep 2021

1. If the authors have adequately addressed your comments raised in a previous round of review and you feel that this manuscript is now acceptable for publication, you may indicate that here to bypass the “Comments to the Author” section, enter your conflict of interest statement in the “Confidential to Editor” section, and submit your "Accept" recommendation.

Reviewer #1: (No Response)

Reviewer #2: All comments have been addressed

2. Is the manuscript technically sounds, and do the data support the conclusions?

Reviewer #1: Partly

Reviewer #2: Partly

RESPONSE: We accepted the comments and tried to incorporate points that clearly stipulated the findings of the study. 

3. Has the statistical analysis been performed appropriately and rigorously? 

Reviewer #1: No

Reviewer #2: I Don't Know

RESPONSE: In the current study, we used the Statistical Package for Social Science Version 24.0 for data analysis. Descriptive analysis (median, percentage, frequencies, and interquartile range) was used to compute demographic characteristics of participants. In addition, bivariable analysis was used to see the significance of the association. Variables that showed strong association (p-value <0.25) in bivariate analysis were entered into multivariable logistic regressions to identify independently associated variables. Multicollinearity was checked by the variance inflation factor (VIF). Statistical significance was declared at a p-value less than 0.05. The significance of association of the variables was described using AOR with a 95% confidence interval.

Unfortunately, there are some editorial errors that occurred during the write up of the result and discussion, such as the report of a female instead of a male in the discussion and exclusion parts. We admitted it and correctly accordingly. Other than this, we did our best and conducted detailed analysis as per the objectives of the study.

4. Have the authors made all data underlying the findings in their manuscript fully available?

Reviewer #1: Yes

Reviewer #2: Yes

5. Is the manuscript presented in an intelligible fashion and written in Standard English?

Reviewer #1: No

Reviewer #2: No

RESPONSE: We have accepted and addressed all the English grammatical errors. 

6. Review Comments to the Author

Reviewer #1: Prevalence of Tobacco Dependence and Associated Factors among Patients with Schizophrenia Attending Their Treatments at Southwest Ethiopia; Hospital-Based Cross-Sectional Study

Reviewer #1:

1. Abstract In the first sentence the word ‘abused’ and ‘abuser’ is not right. There is no criteria for tobacco abuse in DSM-5.

Authors’ response: Accepted and replaced the word ‘abused’ and ‘abuser’ with the word ‘used’ and ‘user’ respectively.

Re-reviewer response -Even though the word is changed the sentence is not written in a correct English grammar.

RESPONSE: Accepted and corrected as “Tobacco smoking is the most commonly used in patients with mental disorders; patients with schizophrenia are the most frequent users”.

Result section of the abstract - Concerning the severity of tobacco dependence, 3.5%, 13.8% and 5% of the respondents report high and very high level of tobacco dependence respectively [ this sentence is not correct, misses the word moderate].

RESPONSE: Accepted and edited accordingly as “concerning the severity of tobacco dependence, 3.5%, 13.8% and 5% of the respondents report moderate, high, and very high level of tobacco dependence respectively”. 

2. The introduction part can be shortened. There are a lot of research findings from all the over the world. It should be summarized to the most relevant researches and high quality evidence.

Authors’ response: we accepted the comments and we have shortened the introduction.

Re-reviewer response – accepted

3. Materials and methods please include a statement on the type of the study design.

Authors’ response: Study design was included in the manuscript

Re-reviewers response – accepted

4. Please include some statements about the interviewer’s level of expertise.

Authors’ response: we have included the statements about the interviewer’s level of expertise under ‘data collection procedures and tools’ subtitle of methods section.

Re-reviewers response – accepted

5. And the instrument that has been used to assess tobacco dependence (Fagerstrom Test for Nicotine Dependence (FTND)), was the instrument used in it’s original language or was it translated?

Authors’ response: we used the translated instrument

Re-reviewers response – accepted

6. If it was translated, please specify the process of translation of this or other instruments.

Authors’ response: the process of translation was described under ‘data quality control’ subtitle of methods section

Re-reviewers response – accepted

7. Please put the reference paper which has validated the SDS instrument in Ethiopia.

Authors’ response: Can be added, because it was validated at Mizan, Ethiopia.

Re-reviewers response – if the validation paper was not published in peer reviewed journal, you cannot say that it was validated.

RESPONSE: The instrument was validated in Mizan, the Southwestern part of Ethiopia. This is the references and we have included also in “Manzar MD, Alamri M, Mohammed S, Khan MAY, Chattu VK, Pandi-Perumal SR, et al. Psychometric properties of the severity of the dependence scale for Khat (SDS-Khat) in polysubstance users. BMC Psychiatry. 2018;18(1):1–8.”

8. Result The 1st paragraph second sentence includes the ‘mean age’.

Authors’ response: we have included

Re-reviewers response – accepted

9. On the next sentences there are repetitions, please correct it.

Authors’ response: we have omitted the repeated sentences

Re-reviewers response – accepted

10. The last sentence the median (IQR) monthly income the IQR is not specified in a range of numbers.

Authors’ response: we accept the comment and corrected accordingly ‘‘the median monthly income of the respondents were 700ETB, which ranges from 100-5000ETB and the interquartile range is 1000.’’

Re-reviewers response – this is still not correct. IQR should be in a range

RESPONSE: IQR was thought to be midspread, the middle 50% of the measure of statistical dispersion, being equal to the difference between 75th and 25th percentiles, or between upper and lower quartiles, and we reported accordingly. Considering the comments we have reported in range.

11. On the bivariate logistic regression,-- it is univariable logistic regression. There is inconsistency of the result from the table 3 and the above paragraph presented about the significant result on univariable and multivariable logistic regression.

Authors’ response: In our analysis part, for univariable analysis we have used a P –value of 0.25 or less as inclusion criteria for the final model so as not to miss important clinical variables. Accordingly male gender, unemployment, being on treatment for 5years, having a history of admission and frequent admission, presence of physical illness, family history of mental illness and being educated above secondary school. However, for the gender, we mistakenly stated and admit to edit to male.

Re-reviewers response – on the paragraph it is still not edited. Table 4 is not written in a consistent manner. For example the ref. value is sometime on the first row and sometimes on the second row. The analysis you have done is not correlation therefore you should not use the words like “correlations”.

RESPOSE: Accepted ad edited accordingly 

12. Discussion It needs further explanation on recommendation and the uniqueness of this study. Authors’ response: we accept and addressed it

Re-reviewers response – not addressed.

RESPONSE: we have included the statements supporting of the first of the discussion and conclusion. 

On the 4th paragraph 3rd line it says “The notion was indicating the role of genetic factors and nicotine dependence in the pathogenesis of schizophrenia” this needs further explanation. Are you saying that nicotine dependence is causing schizophrenia?

RESPONSE: We accepted the comment and corrected the sentence as “the notion was indicating the role of genetic factors in the etiology of smoking behaviour and the high comorbidity between nicotine dependence and schizophrenia” and we are not saying that nicotine dependence is causing schizophrenia. That was an editorial error.

On the conclusion part it says female gender is associated with tobacco dependence and it should male gender.

RESPONSE: we accepted and edited accordingly. That was an editorial error.

In general the whole text needs English language revision. The whole manuscript is written in mundane fashion it should be written in attractive way to the readers.

RESPONSE: we have accepted and addressed all the English language grammar problems. 

Reviewer #2: The authors have addressed all comments, but there are still some issues in the text that must be improved

RESPONSE: We have accepted and addressed all the issues in the text. 

Abstract, Background: „Tobacco smoking is that the most typically employed in patients with mental disorders; among them, patients with schizophrenia area unit the very best users...“ This text is not clear, please, change

RESPONSE: we accepted and changed as “Tobacco smoking is the most commonly used in patients with mental disorders; patients with schizophrenia are the most frequent users”.

Abstract, results: Concerning the severity of tobacco dependence, 3.5%, 13.8% and 5% of the respondents report high and very high level of tobacco dependence respectively-all three percentages need description., i.e., to which two of the three mentioned percentages do belong high and very high level of ND?

RESPONSE: Accepted and edited accordingly above as “concerning the severity of tobacco dependence, 3.5%, 13.8% and 5% of the respondents report moderate, high, and very high level of tobacco dependence respectively”.

Results

The mean age (±SD) of the study participants were 33.7(±7.9) year age..-please, delete „age“ at the end of the sentence

RESPONSE: Accepted and deleted the word “age” from the end of the sentence. 

Table 3. Please, explain what is „frequency “ and percentage“. Does the „frequency“ actually refer to absolute numbers?

RESPONSE: Frequency refers to absolute numbers, while percentage refers to the relative frequency value divided by 100.

While male gender was associated with tobacco dependence in univariable logistic regression, female gender was associated with tobacco dependence in the final regression analysis. It is difficult to understand, so, please, explain this finding, and mention this also in the discussion.

RESPONSE: we accepted and edited accordingly. That was an editorial error. 

Discussion

„The results were also higher compared to the finding of the study from Nigeria (20.4)“-percent?

RESPOSE: Accepted and corrected as 20.4%. That was an editorial error. 

References

References are not complete. Please, check and correct carefully all references.

For example, reference No 10 is lacking five additional authors, and the journal name. References 11 and 12 look very much alike, while the No 12 has no journal name

RESPOSE: 10 corrected, 

7. PLOS authors have the option to publish the peer review history of their article (what does this mean?). If published, this will include your full peer review and any attached files.

Do you want your identity to be public for this peer review? For information about this choice, including consent withdrawal, please see our Privacy Policy.

Reviewer #1: No

Reviewer #2: No(1)

---

## [Decision Letter · Decision Letter 2]

6 Nov 2021

PONE-D-20-31418R2Prevalence of Tobacco Dependence and Associated Factors among Patients with Schizophrenia Attending Their Treatments at Southwest Ethiopia ; Hospital-Based Cross-Sectional StudyPLOS ONE

Dear Dr. Desalegn,

Thank you for submitting your manuscript to PLOS ONE. After careful consideration, we feel that it has merit but does not fully meet PLOS ONE’s publication criteria as it currently stands. Therefore, we invite you to submit a revised version of the manuscript that addresses the points raised during the review process.

 The reviewer again addressed several minor concerns about your manuscript. Please revise your manuscript carefully.

We look forward to receiving your revised manuscript.

Kind regards,

Kenji Hashimoto, PhD

Academic Editor

PLOS ONE

Journal Requirements:

Reviewers' comments:

Reviewer's Responses to Questions

**Comments to the Author**

1. If the authors have adequately addressed your comments raised in a previous round of review and you feel that this manuscript is now acceptable for publication, you may indicate that here to bypass the “Comments to the Author” section, enter your conflict of interest statement in the “Confidential to Editor” section, and submit your "Accept" recommendation.

Reviewer #2: (No Response)

2. Is the manuscript technically sound, and do the data support the conclusions?

Reviewer #2: Yes

3. Has the statistical analysis been performed appropriately and rigorously? 

Reviewer #2: Yes

4. Have the authors made all data underlying the findings in their manuscript fully available?

Reviewer #2: Yes

5. Is the manuscript presented in an intelligible fashion and written in standard English?

Reviewer #2: Yes

6. Review Comments to the Author

Reviewer #2: Authors have addressed the comments

However, there are some additional remarks:

In tables 1, 2 and 3, frequency should be replaced with numbers

References should all be edited in a uniform way, strictly according to the journal policy. For example, references no 5,6,8,9, 13,14,15,16,19, etc, have no journal name. Reference no 10 has full author first names, some references have name of the month, for example the ref no 20, which is missing journal volume and pages. Please, correct all references!

English grammar has to be checked and corrected

7. PLOS authors have the option to publish the peer review history of their article (what does this mean?). If published, this will include your full peer review and any attached files.

Reviewer #2: No

---

## [Author Response · Author response to Decision Letter 2]

17 Nov 2021

RESPONSE TO EDITOR AND REVIEWER’

Journal Requirements:

RESPONSE: We have accepted and corrected all the references according to the journal policy. 

Reviewers' comments:

Reviewer's Responses to Questions

Comments to the Author

1. If the authors have adequately addressed your comments raised in a previous round of review and you feel that this manuscript is now acceptable for publication, you may indicate that here to bypass the “Comments to the Author” section, enter your conflict of interest statement in the “Confidential to Editor” section, and submit your "Accept" recommendation.

Reviewer #2: (No Response)

2. Is the manuscript technically sound, and do the data support the conclusions?

Reviewer #2: Yes

3. Has the statistical analysis been performed appropriately and rigorously? 

Reviewer #2: Yes

4. Have the authors made all data underlying the findings in their manuscript fully available?

Reviewer #2: Yes

5. Is the manuscript presented in an intelligible fashion and written in standard English?

Reviewer #2: Yes

6. Review Comments to the Author

Reviewer #2: Authors have addressed the comments

However, there are some additional remarks:

In tables 1, 2 and 3, frequency should be replaced with numbers

RESPOSE: Accepted and we have replaced frequency with numbers accordingly

References should all be edited in a uniform way, strictly according to the journal policy. For example, references no 5,6,8,9, 13,14,15,16,19, etc, have no journal name. Reference no 10 has full author first names, some references have name of the month, for example the ref no 20, which is missing journal volume and pages. Please, correct all references!

RESPONSE: We have accepted and corrected all the references according to the journal policy. 

English grammar has to be checked and corrected

RESPONSE: we have accepted and addressed all the English language grammar problems. 

7. PLOS authors have the option to publish the peer review history of their article (what does this mean?). If published, this will include your full peer review and any attached files.

Do you want your identity to be public for this peer review? For information about this choice, including consent withdrawal, please see our Privacy Policy.

Reviewer #2: No

---

## [Decision Letter · Decision Letter 3]

29 Nov 2021

Prevalence of Tobacco Dependence and Associated Factors among Patients with Schizophrenia Attending Their Treatments at Southwest Ethiopia ; Hospital-Based Cross-Sectional Study

PONE-D-20-31418R3

Dear Dr. Desalegn,

We’re pleased to inform you that your manuscript has been judged scientifically suitable for publication and will be formally accepted for publication once it meets all outstanding technical requirements.

Kind regards,

Kenji Hashimoto, PhD

Section Editor

PLOS ONE

Additional Editor Comments (optional):

Reviewers' comments:

Reviewer's Responses to Questions

**Comments to the Author**

1. If the authors have adequately addressed your comments raised in a previous round of review and you feel that this manuscript is now acceptable for publication, you may indicate that here to bypass the “Comments to the Author” section, enter your conflict of interest statement in the “Confidential to Editor” section, and submit your "Accept" recommendation.

Reviewer #2: All comments have been addressed

2. Is the manuscript technically sound, and do the data support the conclusions?

Reviewer #2: Yes

3. Has the statistical analysis been performed appropriately and rigorously? 

Reviewer #2: Yes

4. Have the authors made all data underlying the findings in their manuscript fully available?

Reviewer #2: Yes

5. Is the manuscript presented in an intelligible fashion and written in standard English?

Reviewer #2: Yes

6. Review Comments to the Author

Reviewer #2: References are again not presented in a uniform way. For example, some do mention the month, such as: Schizophrenia research. 2005 Jul 15;76(2-3):135-57., while others don't, for example: Neuropsychiatr Dis Treat. 2018;14:1535–43.

7. PLOS authors have the option to publish the peer review history of their article (what does this mean?). If published, this will include your full peer review and any attached files.

Reviewer #2: No

---

## [Editor Report · Acceptance letter]

6 Dec 2021

PONE-D-20-31418R3 

Prevalence of tobacco dependence and associated factors among patients with schizophrenia attending their treatments at southwest Ethiopia; Hospital-based cross-sectional study 

Dear Dr. Desalegn:

I'm pleased to inform you that your manuscript has been deemed suitable for publication in PLOS ONE. Congratulations! Your manuscript is now with our production department. 

Kind regards, 

on behalf of

Prof. Kenji Hashimoto 

Section Editor

PLOS ONE